

# AR2-D2:
# Training a Robot Without a Robot

**Jiafei Duan** [1]   **Yi Ru Wang** [1]   **Mohit Shridhar** [1]   **Dieter Fox** [1,2]   **Ranjay Krishna** [1,3]
[1]University of Washington   [2]NVIDIA   [3]Allen Institute for Artificial Intelligence
{duanj1, yiruwang, mshr, fox, ranjay}@cs.washington.edu

[www.ar2d2.site](www.ar2d2.site)

**Abstract:** Diligently gathered human demonstrations serve as the unsung heroes empowering the progression of robot learning. Today, demonstrations are collected by training people to use specialized controllers, which (tele-)operate robots to manipulate a small number of objects. By contrast, we introduce AR2-D2: a system for collecting demonstrations which (1) does not require people with specialized training, (2) does not require any real robots during data collection, and therefore, (3) enables manipulation of diverse objects with a real robot. AR2-D2 is a framework in the form of an iOS app that people can use to record a video of themselves manipulating any object while simultaneously capturing essential data modalities for training a real robot. We show that data collected via our system enables the training of behavior cloning agents in manipulating real objects. Our experiments further show that training with our AR data is as effective as training with real-world robot demonstrations. Moreover, our user study indicates that users find AR2-D2 intuitive to use and require no training in contrast to four other frequently employed methods for collecting robot demonstrations.

**Keywords:** Demonstration Collection, Imitation Learning, Augmented Reality

## 1 Introduction

Manually curated datasets are often the inglorious heroes of many large-scale machine learning projects [1, 2, 3]; this is especially true in robotics, where human-generated datasets of robot demonstrations are indispensable [4, 5] especially with recent success in robot learning via imitation learning [6, 7, 8, 9] of these demonstration data. For example, one recent effort collected $\sim 130k$ robot demonstrations, with a fleet of 13 robots over the course of 17 months [10]. As a result, researchers have spent considerable effort developing various mechanisms for demonstration collection. One popular option for collecting robot demonstrations is through kinesthetic-teaching, where a person guides a robot through a desired trajectory [11]. Although intuitive, this mechanism can be tedious and slow [12]. Alternatively, teleoperation with various controllers has become popular: using a keyboard and mouse [13, 14], a video game controller [15], a 3D-mouse [16, 6], special purpose master-slave interfaces [17, 18], and even virtual reality (VR) controllers [19, 20, 21].

Despite all these demonstration collection efforts, there are three key challenges limiting robot data collection. First, people need to be trained to produce useful demonstrations: kinesthetic methods are labor-intensive while teleoperation methods require learning specialized controllers. Second, the ability to parallelize data collection is limited by how many—often expensive—robots are available. Third, robots are usually bulky and locked within a laboratory, reducing their exposure to a handful of nearby objects. Lastly, (tele)-operation in simulation has the potential to scale efficiently without real robot hardware, but addressing the sim2real gap and limited variety of trainable tasks in simulation environments are challenges to overcome.

7th Conference on Robot Learning (CoRL 2023), Atlanta, USA.

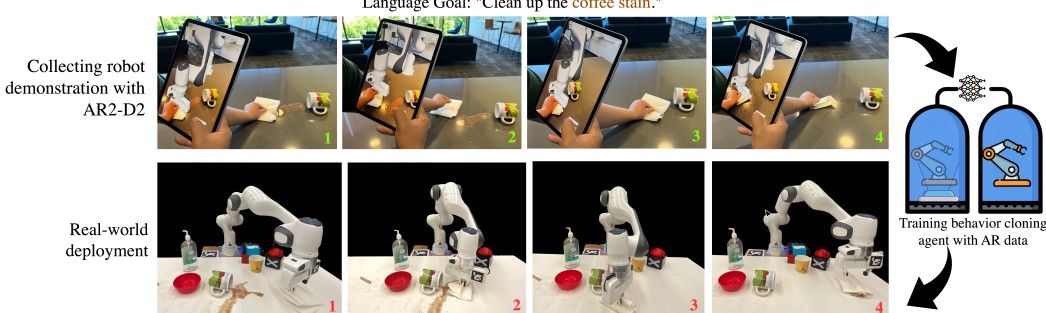

Figure 1. **AR2-D2** collects robot demonstrations without needing a real robot. (**Top left**) Using AR2-D2, the user captures a video manipulating an object with their arm. AR2-D2 projects an operational URDF (unified robotics description format) of an AR Franka Panda robot arm into a physical environment. It uses a hand-pose tracking algorithm to move the AR robot's end effector to align with and mirror the 6D pose of the human hand. (**Right**) With this video demonstration, we train a perceiver-actor agent and (**Bottom left**) deploy the agent on a real-world robot to demonstrate its ability to learn from AR demonstrations.

We introduce AR2-D2: a system for collecting robot demonstrations that (1) does not require people to have specialized training, (2) does not require any real robots during data collection, and therefore, (3) enables manipulation of diverse objects with a real robot. AR2-D2 is a framework in the form of an iOS app that enables users to record a video of themselves manipulating any object. Once the video is captured, AR2-D2 uses the iOS depth sensor to place an AR robot in the scene and uses a motion planner to produce a trajectory where it appears as if the AR robot manipulates the object (Figure 1). Manipulating objects and recording a video is so intuitive that users do not need any training to use AR2-D2. Our system completely removes the need for a real robot during demonstration collection, allowing data collection to potentially parallelize without being limited by expensive real robots. Finally, since videos can be captured anywhere, AR2-D2 re-situates demonstration collection out of the laboratory; users can take videos anywhere, making it easy to collect demonstrations involving manipulation of diverse objects. Furthermore, unlike collecting visual observations of human activities, our approach uses AR projection during robot demonstration collection to provide constant feedback on the robot's pose and physical constraints in the given environment while performing the task.

Our experiments show that AR2-D2's AR demonstrations can effectively train a real robot to manipulate real-world objects. We use AR2-D2 to collect robot demonstrations cross three manipulation tasks (*press*, *pick-up* and *push*) on 9 personalized objects. These personalized objects are uniquely shaped, sized, or textured items designed to meet the specific needs or functionalities of individual users within their personalized environments. We collect and use as few as five effortlessly collected demonstrations to train a behavior cloning agent [6]. This trained agent needs to be finetuned for 3,000 iterations (which is equivalent to less than 10 minutes of training) on a dummy real-world task to overcome the sim-to-real gap; Once finetuned, a real robot is capable of manipulating real objects even though that object was only encountered by the AR robot. This AR-trained agent performs comparable to agents trained with real-world tele-operated demonstrations (specifically the PerAct demonstration collection [6]).

We assess AR2-D2's usability through a within-subjects user study (N=10). For the user study, participants are asked to provide demonstrations for two standard manipulation tasks: *pick-up* and *stacking*. Besides AR2-D2, users collect demonstrations using four alternative methods, including keyboard and mouse [22], VR controller [23], 3D-mouse [22], and kinesthetic-teaching [24]. Results suggest AR2-D2 is intuitive, requires no training, and enables quick demonstration production, comparable to kinesthetic teaching and faster than other methods. AR2-D2 paves the way for democratizing robot training: an estimated 1.36 billion[1] iPhone users could create personalized manipulation data to train real robots for their household objects.

---

[1]Source: `https://www.bankmycell.com/blog/number-of-iphone-users`

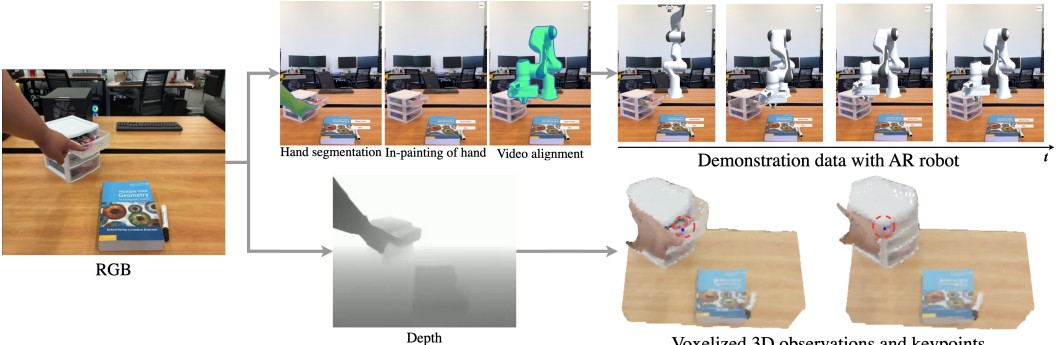

*Figure 2.* **AR2-D2 collection process**. **(Left)** Once the user records themselves manipulating an object, AR2-D2 extracts the following information: 6D hand pose, hand state, RGB frames and depth estimations. We replace the hand with an AR robot, aligning its motions to align its end effector with the hand's. **(Right)** We create a 3D voxelized observation over time from the extracted keyframes. This 3D representation is used to train a PERACT [6] agent. We also use the generated video to train an image-conditioned BC agent [6].

## 2 Related work

**Demonstration collection methods.** There are several conventional methods available for gathering robot demonstrations. One popular approach involves kinesthetically controlling the robot to follow a desired trajectory; the generated robot trajectories showcase the behavior required to accomplish a specific task [11]. Teleoperation techniques [25, 26, 27] have also been a popular collection process, using various user interfaces including keyboard and mouse [13, 14], a video game controller [15], 3D-mouse [16, 6], mobile phones [28], special purpose master-slave interfaces [17, 18],virtual reality controllers [19, 20, 21, 29] and using human videos and extracts visual priors to project them into a simple set of robot primitives for collect robot demonstrations [30, 31, 32, 33, 34]. However, all of these methods require a real physical robot to be controlled, bottle-necking demonstration collection by how many robots are available and limited to the laboratories that house these robots.

**Demonstrations for behavior cloning.** Recently, robot demonstrations are primarily used as training data for imitation learning, which has pioneered a paradigm shift in robot training. Offline behavior cloning from robot demonstrations is currently the de-facto imitation learning paradigm [35]. These demonstrations are collected either in simulation or through human control using a real robot in the real world [36, 37]. For example, Task and Motion Planning (TAMP) uses expert task planners to create large-scale simulation demonstrations [36]. Meanwhile in the real-world, users employ techniques such as teleportation or vision-based guidance are used to create demonstrations [20, 38, 39, 7]. Recent methods have also begun developing specialized hardware to streamline demonstration collection. For example, a low-cost handheld device featuring a plastic grabber tool outfitted with an RGB-D camera and a servo can control the binary opening and closing of a grabber's fingers [40]. By contrast, our real-world data collection approach requires no teleoperation hardware [28], no simulators [41], and most importantly, no real robots [42]. All we need is an iPhone camera to record users manipulating objects with their hands.

## 3 The AR2-D2 system

We introduce AR2-D2, a system for collecting robot demonstrations without requiring a physical robot. In this section, we describe AR2-D2's features, its supported data collection procedure, its implementation details.

### 3.1 AR2-D2 system features

AR2-D2 contributes the following features:

**No need for a physical robot.** In traditional robotics research, obtaining demonstrations often involves operating a physical robot [38, 39, 20, 40]. AR2-D2 presents a new paradigm for collecting

demonstrations; it forgoes access to a real robot, enabling users to collect high-quality demonstration data from anywhere with only their mobile devices.

**Real-time control of AR robots in the real-world.** AR2-D2 leverages LiDAR sensors, which today are ubiquitous in iPhones and other smartphones to estimate the 3D layout in front of the camera to project an AR robot. LiDAR helps the AR robot obey physical and visual realism. Users can control the AR robot in one of three supported interactions: by pointing at 3D points that the robot's end-effector should move to, by using the iPhone's GUI control, or through AR kinesthetic control (see appendix). The projected robot's motions are tightly coupled with the real-world environment, and receives feedback upon collisions with real-world objects.

**Real-time visualization of task feasibility.** AR2-D2 simplifies the demonstration collection by asking users to specify key-points that the robot end-effector should move to in order to complete a task. Once each key-point is specified, AR2-D2 visualizes the AR robot's motion, moving its end-effector from its current position to the new key-pose. This real-time feedback enables users to assess the feasibility and accuracy of the specified key-point and revise their selections if necessary.

## 3.2 Design and implementation

AR2-D2's design and implementation consists of two primary components (Figure 2). The first component is a phone application that projects AR robots into the real-world, allowing users to interact with physical objects and the AR robot. The second component convert collected videos into a format that can be used to train different behavior cloning agents, which can later be deployed on real robots.

**The phone application.** We designed AR2-D2 as an iOS application, which can be run on an iPhone or iPad. Since modern mobile devices and tablets come equipped with in-built LiDAR, we can effectively place AR robots into the real world. The phone application is developed atop the Unity Engine and the AR Foundation kit. The application receives camera sensor data, including camera intrinsic and extrinsic values, and depth directly from the mobile device's built-in functionality. The AR Foundation kit enables the projection of the Franka Panda robot arm's URDF into the physical space. To determine user's 2D hand pose, we utilize Apple's human pose detection algorithm. This, together with the depth map is used to reconstruct the 3D pose of the human hand. By continuously tracking the hand pose at a rate of 30 frames per second, we can mimic the pose with the AR robot's end-effector.

**Training data creation.** Given language instructions for a task (e.g., "Pick up the plastic bowl"), we hire users to generate demonstrations using AR2-D2. From the user-generated demonstration video, we create training data to train and deploy on a real robot. To make this training data, we convert the video to show an AR robot manipulating the object. We remove the human hand with Segment-Anything [43] and fill the gap left behind by the missing hand with a video in-painting technique, E2FGVI [44]. Finally, we produce a video with the AR robot arm moving to the user's hand's key-points. This processed video makes it look like an AR robot arm manipulated the object and can be used as training data for visual-based imitation learning [45]. Additionally, with access to the scene's depth estimation, we can create a 3D voxelized representation of the scene to train agents like Perceiver-Actor.(PERACT) [6].

## 4 Evaluating AR2-D2 with real users

To evaluate AR2-D2's efficacy, we conduct an extensive within-subjects user study ($N = 10$) across 5 demonstration collection techniques for 2 tasks. Participants demonstrated each task 3 times with each technique, resulting in a total of 300 collected demonstrations. Participants were locally hired; they were aged between 23 and 30.

**Baselines collection techniques.** In order to compare how effectively real participants create demonstrations with AR2-D2, we also ask them to use 4 other baseline collection techniques. Two collection techniques utilize real robots in the real-world and two control simulation robots. In sim-

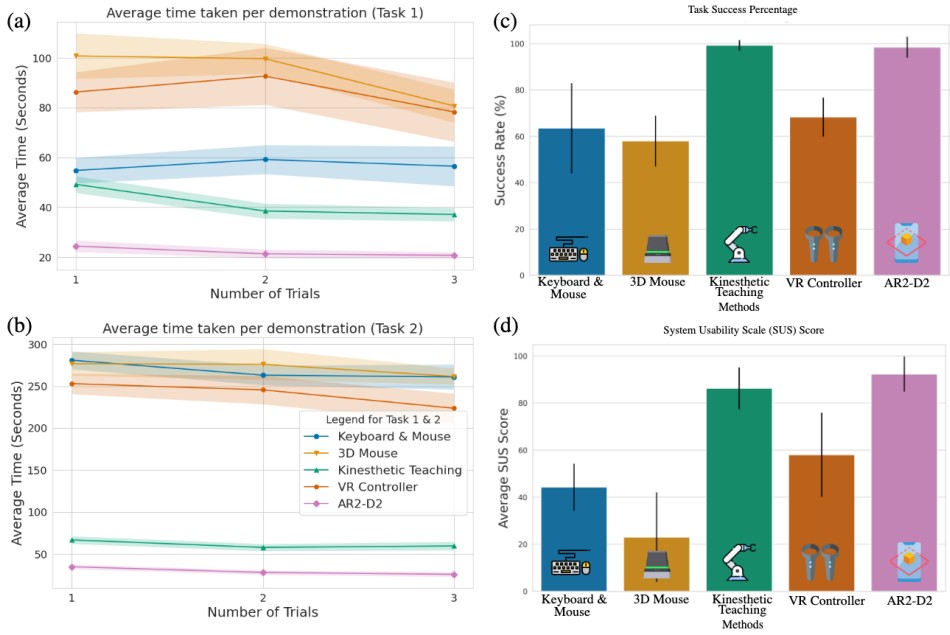

Figure 3. **Evaluating AR2-D2 with real users.** We conduct an extensive within-subjects user study, comparing AR2-D2 against 4 alternative collection techniques: keyboard & mouse, 3D mouse (6-DoF), kinesthetic teaching, and HTC Hive controller. (a, b) We find that participants spend significantly less time (with an average of 22.1 and 29.5 seconds across the two tasks) using our system than others versus the next best (kinesthetic teaching with an average of 41.6 and 61.4 seconds). (c, d) We show that participants are able to successfully collect a demonstration with the same rate of success using our system as kinesthetic teaching, both of which have significantly higher success rate versus others.

ulation, participants control a simulated Franka Panda with either keyboard and mouse or with a 3D Space Mouse. Using the keyboard and mouse, users can manipulate the 6D end-effector of the simulated robot within the Isaac Sim environment, utilizing ORBIT [22]. The 3D Space Mouse is a joystick capable of simultaneous translation and rotation along the (x, y, z) axes; it operates within the same environment as the keyboard. In the real world, participants use kinesthetic teaching or an HTC Vive VR controller. Kinesthetic teaching allows participants to manipulate a real Franka Panda, using its default zero-gravity feature. The demonstration collection interface using the HTC Vive VR controller was developed in a recent paper and enables teleoperation of the robot [29].

**Study protocol.** Each participant was tasked with collecting demonstrations for two specific tasks: *picking* and a *stacking* (Figure 3). Participants were asked to provide demonstrations for each task across 3 trials, with 3 attempts allowed per trial. We imposed a time constraint are each trial: 3 minute limit for the *picking* and a 5 minute limit for *stacking*. After all the data was collected, participants filled out a system usability scale (SUS) survey.

**Measured variables.** We evaluate the different data collection techniques using two metrics. First, we measure average data collection time (in seconds). Lower values are better because it implies that participants are able to collect demonstrations quicker. Second, we measure task success rate, which calculates the percentage of trials that lead to a successful demonstration.

**Results.** We show that participants using AR2-D2 are both significantly faster (Figure 3 (a, b)) and as likely (Figure 3 (c, d)) to collect a successful demonstration as kinesthetic teaching. In comparison with kinesthetic teaching, which has an average task completion time of 41.6 and 61.4 seconds for task 1 and 2 respectively, our method exhibits a substantial reduction in time with only 22.1 and 29.5 seconds for both tasks respectively. Furthermore, the t-tests for task 1 and task 2 yielded t-statistics of $t_1 = 6.194$ and $t_2 = 6.199$, with p-values of $p_1 = 7.587 \times 10^{-6}$ and $p_2 = 7.514 \times 10^{-6}$ respectively. Hence, we could confidently say that there is a statistically significant difference between kinesthetic teaching and our approach, with kinesthetic teaching having, on average, significantly longer timings compare to ours. This concludes that our method is capable of collecting robot demonstrations faster than the traditionally favored kinesthetic teaching.

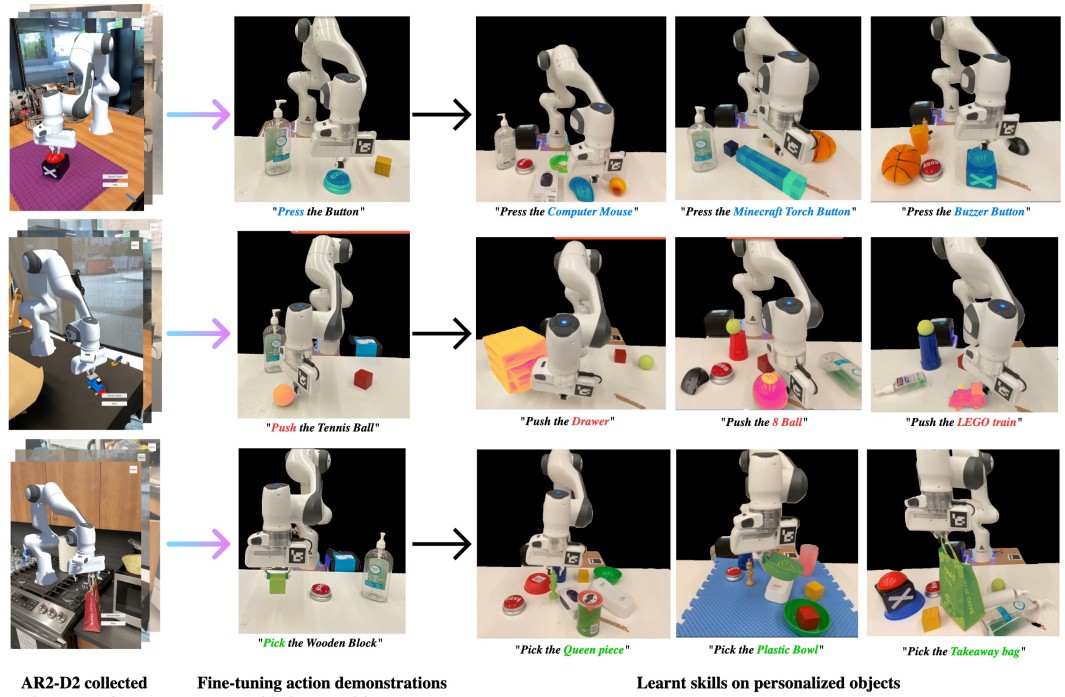

| AR2-D2 collected demonstrations | Fine-tuning action demonstrations with dummy objects | Learnt skills on personalized objects |

*Figure 4.* **Evaluating AR2-D2 data by training a real robot to manipulate real objects**. We employ AR2-D2 as a tool for gathering a diverse array of manipulations encompassing three fundamental actions, involving a wide variety of customized objects. These manipulations range from performing precise actions such as pressing a computer mouse or a Minecraft torch button at specific locations, to pushing small LEGO train toys towards table-sized drawers, and even encompassing the ability to pick up objects varying from chess pieces to takeaway bags. By leveraging a limited number of real-world action demonstrations conducted with random dummy objects and fine-tuning for 3,000 iterations which is equivalent to 10 minutes of training, we have achieved the capacity to apply the PerAct framework in manipulating all these personalized objects with broad generalization.

| Task | Press (*Succ.%*) | | | Push (*Succ.%*) | | | Pick up (*Succ.%*) | | |
| Personalized object | Computer mouse | Minecraft torch | Buzzer | LEGO train | 8 ball | Drawer | Queen piece | Plastic bowl | Takeaway bag |
|---|---|---|---|---|---|---|---|---|---|
| Simulation | 13.3 | 6.7 | 30.0 | 13.3 | 20.0 | 3.3 | 3.3 | 20.0 | 16.7 |
| VR interface (w/o personalized objects) | 3.3 | 6.7 | 16.7 | 13.3 | 10.0 | 3.3 | 0.0 | 16.7 | 13.3 |
| VR interface (with personalized objects) | 60.0 | 63.3 | 83.3 | 30.0 | 70.0 | 40.0 | 46.7 | 56.7 | 60.0 |
| AR2-D2 (Ours) | 56.7 | 53.3 | 73.3 | 50.0 | 55.7 | 23.3 | 46.7 | 53.3 | 63.3 |

*Table 1.* **Task test results**. We utilized AR2-D2 to collect demonstrations and train BC agents for real robot deployment. Our observations revealed comparable results between our data collection approach and alternative methods. Success rates (mean %) of the foundational skills tested on personalized objects collected via AR2-D2. For each skill, we evaluated it across ten different sets of distractors with the target object and repeated thrice for consistency. The result has shown that our data collection approach with minimal fine-tuning achieves comparable results to real-world data collected on these personalized objects via PerAct's VR interface with a physical robot.

We find that participants using AR2-D2 are fast from the get-go (Figure 3(a, b)). Participants are consistently faster when collecting demonstrations from the very first trial. This consistency is reflected in the relatively low standard deviation values of 5.75 and 8.89 seconds for the two tasks across participants. In contrast, the next quickest contender, kinesthetic teaching, exhibits a standard deviation of 9.62 and 14.02. Additionally, users have indicated a higher preference for our system in the SUS survey [46] (Figure 3 (d)). Our method garners a similar level of user preference as kinesthetic-teaching, which necessitates a physical robot, with a mere ±6% difference in SUS scores between the two techniques.

## 5 Evaluating AR2-D2 with a real robot deployment

With AR2-D2, we collect demonstrations and train behavior cloning agents for deployment on a real robot. Here, we present our experimental setup and three key takeaways. First, we validate that AR2-D2 demonstrations can train a real robot to manipulate personalized objects without access to a physical robot. Second, the agent trained using AR2-D2's demonstrations perform on par with training on real robot demonstrations. Third, AR2-D2's demonstrations can enable learning policies from both image as well as voxelized inputs.

**AR2-D2 demonstration collection.** We collect AR2-D2 demonstrations on a set of personalized objects, and demonstrate that a policy trained on this data executes on a real Franka Panda robot. We gather demonstrations centered around three common robotics tasks: {*press, push, pick up*}. The three tasks are delineated as follows: pressing down on the targeted object, pushing the targeted object across a surface, and picking up the targeted object. For each task, we collect five demonstrations using three different objects, which vary in color, size, geometry, texture, and even functionality (see Figure 4).

**Behaviour cloning.** We use Perciver-Actor (PERACT) [6] to train a transformer-based behavior cloning policy. PERACT takes a 3D voxel observation and a language goal $(v, l)$ as input and produces outputs for translation, rotation, and gripper state of the end-effector. These outputs, with a motion planner, enable the execution of the task specified by the language goal.

**Training procedure.** Following existing work [6], we train an individual agent for each task. We train an agent for 30k iterations per set of demonstrations. We then freeze the backbone of the PERACT architecture and finetune the rest using the set of VR (using HTC Hive) demonstrations on dummy objects. This fine-tuning process spans an additional 3k iterations, equivalent to approximately 10 minutes of wall clock training. Fine-tuning allows us to close the domain gap resulting from differences in depth cameras between the Kinect V2 used by PERACT and the iPhone/iPad depth camera used by AR2-D2.

**Finetuning demonstration collection.** Finetuning demonstrations are collected using the VR interface from PerAct [6]. It involves using a VR handset to guide the real-robot to desired end-effector poses. We collect 5 demonstrations for each task using three dummy objects: {*red button, yellow block, tennis ball*}, corresponding to the three tasks, respectively. These specific objects are only used for fine-tuning and not used during testing. We also ablate the agent's performance without finetuning.

| Task | 2D data (Image-BC) | 3D data (PerAct) |
|------|------|------|
| Press the buzzer from the side | 0.00% | 40.00% |
| Pick up the queen piece | 6.67% | 33.34% |
| Press the computer mouse | 6.67% | 40.00% |

*Table 2.* **Training with Different Data Modalities.** AR2-D2 is capable of offering diverse data modalities to facilitate training BC models, such as Image-BC for 2D data and PerAct for 3D data. We assess these disparate data modalities, gathered via AR2-D2, across three distinct tasks using Image-BC for 2D data and PerAct for 3D data, conducted without any form of fine-tuning

**Testing procedure.** We evaluate the trained policies' ability to manipulate personalized objects in the real world. The personalized objects are comprised of distinctly different objects from the AR2-D2-enabled real-world demonstrations. Each test environment is infected with ten different distractor objects. We repeat run inference three times for each environment setup and average their performance.

**Baseline collection techniques.** We compare AR2-D2's demonstrations against two alternative techniques: real-world and simulation data collection. We finetune all methods using the same set of finetuning demonstrations on dummy objects. *Real world* data collection uses a VR controller interface to capture the training demonstra-

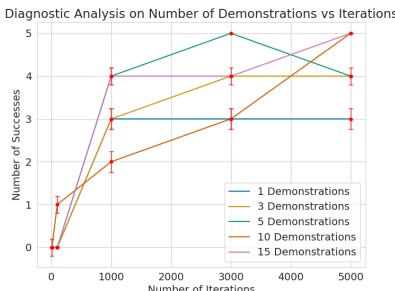

*Figure 5.* **Analysis on Fine-tuning.** We conducted a diagnostic analysis to determine the optimal number of iterations and demonstrations required. By varying the number of demonstrations and iterations for fine-tuning, we found that using 5 demonstrations and 3,000 iterations yielded the best results.

tions [6]. Real-world demonstrations are collected with and without the personalized objects (see Table 1). *Simulation* demonstrations use RLBench and its key-frame point extraction technique, accompanied by motion planning to generate each demonstration [41]. We implemented domain randomization to introduce texture variations, aiding in the transfer from simulation to the real world.

## 5.1 Results

Table 1 reports success rates of all the nine personalized objects across three tasks with demonstrations from real-world, from simulation, and from AR2-D2.

**AR2-D2 demonstrations yields useful representation for training a real-robot.** In general, AR2-D2's data outperforms policies trained using simulation of real-world (without personalized objects). In fact, in one case, we outperform PERACT's real world data collection (without personalized objects) by a large margin of $53.4\%$. These findings highlight the significance of our approach, which facilitates access to collecting demonstrations with such personalized objects which might not be available in the laboratory that houses the robot. This capability to produce training data with personalized objects is particularly important since behavior cloning agents perform better when their training exposures them to the objects they are expected to manipulate.

**AR2-D2 demonstrations train policies as accurately as demonstrations collected from real robots.** Referencing Table 1, it is evident that even when real-world data collection is trained with personalized objects during the demonstration data collection phase, our method delivers comparable results. Remarkably, our system's data even surpasses the real-robot collection data in tasks such as pushing the LEGO train and picking up the paper bag. While for the remaining personalized objects, our approach maintains a $\leq 14.3\%$ gap across the three foundational skills. The t-test results, with a calculated *t-value* of 0.547 and a *p-value* of 0.592, indicating that there is no statistical significance in the observed difference between the two methods.

## 5.2 Ablations

**Analysis on Fine-tuning.** We investigate how many finetuning demonstrations ($\{1, 3, 5, 10, 15\}$) on dummy objects and how many training iterations ($\{0, 1000, 2000, 3000, 4000, 5000\}$) are required to maximize the agent's performance. These ablations pretrain the policy using 5 AR2-D2 demonstrations of the "mouse" pressing task trained for 30k training iterations. Each ablation is tested on 1 real-world scene with the computer mouse but we evaluated it across 5 trials with varying target object poses and placement. We find that 5 fine-tuning demonstrations trained for 3k iterations (equivalent to 10 minutes of training) yields the most effective outcome (see Figure 5).

**Training with voxelized inputs is better than using 2D inputs.** AR2-D2 demonstrations store 2D image and 3D depth data, facilitating training of image-based behavior cloning (Image-BC) and 3D voxelized methods (PERACT [6]). With fixed camera calibration offset and no finetuning during training, 3D-input agents outperform 2D counterparts (refer to Table 2 and supplementary).

# 6 Limitations and Conclusion

**Limitations.** Our research does present certain limitations. Firstly, due to the inherent characteristics of our method, it proves challenging to validate experimental outcomes via simulation. Consequently, the verification relies on real-world assessments, which, despite our extensive multi-trial evaluations using varied layouts, cannot completely encompass all conceivable scenarios. Secondly, while our user-study participant count mirrors the standards set by RoboTurk [28], we acknowledge that a larger participant pool might have enhanced the statistical significance of the performance results across various methods. Lastly, due to the disparity between the camera sensors and the domain gap, there is still a need for fine-tuning to match the performance of real data. Nevertheless, future work can explore better approaches to further bridge this domain gap either through better data augmentation techniques or hardware such as Apple's AR head-mounted display.

**Conclusion.** We present AR2-D2, an intuitive robot demonstration collection system that enables the collection of quality robot demonstrations for diverse objects without the need for any real robots or the need to train people before use. Our results highlight the effectiveness of this approach, showing that as few as five AR demonstrations suffice to train a real-world robot to manipulate personalized objects. Our extensive real-world experiments further confirmed that AR2-D2's AR data is on par with training using real-world demonstrations. Moreover, through our comprehensive user-study, it revealed that users found our method intuitive and easy to use, requiring no prior training, setting it apart from traditional collection methods. Finally, AR2-D2 paves the way towards democratizing robot training by enabling any individual to gather significant robot training data for manipulating their personalized objects at any place and time.

**Acknowledgments**

We thank the members of the Robotics State Estimation lab and Krishna's group for the helpful discussions and feedback on the paper. Jiafei Duan is supported by the National Science Scholarship from The Agency for Science, Technology and Research (A*STAR), Singapore.

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
