# OpenReview forum: "AR2-D2: Training a Robot Without a Robot"
_robot-learning.org/CoRL/2023/Conference — CoRL 2023 Poster_

### Official Review · Reviewer_UipD · 2023-06-28

**Confidence:** 4
**Originality:** Very Good
**Technical Quality:** Very Good
**Clarity Of Presentation:** Excellent
**Impact:** 4

**Recommendation:**

Strong Accept: I recommend accepting the paper and will argue for my recommendation even if other reviewers hold a different opinion.

**Review:**

**Strength**:

1. The proposed approach utilizes augmented reality to project an AR Franka robot into a physical environment, freeing human demonstrators from the need for teleoperation of the robot. This has the potential to create a scalable method for generating robotics data in real-world settings.
2. The manuscript provides evidence of comparable results between the proposed AR-based method and human teleoperation using VR and kinesthetic teaching of the robot.
3. The core method, which involves AR data collection without a physical robot, fine-tuning using a few real robot demonstrations, and deployment, is straightforward and easy to understand.

**Weakness**:

1. The requirement for fine-tuning data still necessitates physical robot demonstrations, indicating a significant AR-to-real gap. However, the reviewer believes that this issue can be addressed in future work.
2. Some details are missing, as mentioned in the questions for the rebuttal section, which the reviewer finds confusing in terms of the methodology.


**Quality Of The Limitations Section:**

Limitations are addressed clearly

**Questions For Rebuttal:**

1. Line 141-143: “This final video processed video makes it appear as if an AR robot arm manipulated the real-world object”: How are the gripper action parametrized in the data? Does the user have to specify whether the gripper should be open or closed when arriving at each key points?
2. Line 140-141: “ Finally, we produce a video with the AR robot arm moving to the keypoints identified by the user’s hand.”: Since depth data is collected, is there in-painting of user’s hand in-depth data? Also, it seems that the robot is placed only in the video-image space, but not in the depth observation space. Is this the source for the AR2Real gap?
3. Does the setup assume that there’s minimal hand occlusion? For example, if we have a task that requires bimanual motions, will the same setup work?
4. Does the AR setup assume a fixed camera pose? Is this a source of concern for transferring a policy from AR2Real?


**Robotics Focus:**

Sufficient demonstration on hardware

**Summary Of Paper:**

The manuscript proposes an approach to scale data in robotics, designed to collect human demonstrations for robot learning using augmented reality. It allows users to record videos of themselves manipulating objects and then modify the captured keyframes, replacing human hand manipulation with robot manipulation of the same object. The manuscript presents experimental results that utilize the system to train an imitation learning agent to manipulate real objects and benchmark the system's usability against traditional methods of collecting human demonstrations (such as kinesthetic teaching or AR controllers).

**Summary Of Recommendation:**

The reviewer recommends strong acceptance. Thanks reviewer again for the timely response to the review. The manuscript introduces a novel approach for collecting robot demonstrations in real-world environments using augmented reality. Users can record demonstrations through human hand movements, and an augmented reality Franka robot mimics the recorded motions. It should be noted that although the title of the manuscript suggests training a robot without a physical robot, there is still an AR-to-real gap that necessitates fine-tuning the policy model with some actual robot demonstrations

---

### Official Review · Reviewer_sq69 · 2023-07-15

**Confidence:** 4
**Originality:** Very Good
**Technical Quality:** Good
**Clarity Of Presentation:** Very Good
**Impact:** 3

**Recommendation:**

Weak Accept: I recommend accepting the paper, but will not argue for my recommendation if the majority of other reviewers have a different opinion.

**Review:**

Strengths
- The proposed approach is attractive, as the authors have highlighted, in scaling up robot learning by removing the requirement of a real robot for data collection

- The authors have demonstrated competitive model performance using the proposed method, while also presented convincing results in terms of usability and collection speed.

Weaknesses
- During training, the model still requires fine tuning using teleop data, since the data presents a hybrid of a real environment with a simulated robot.

- The fidelity of the physical interaction between the object and the simulated arm depends on the accuracy of end effector rotation/position estimation. It is somewhat unclear whether interactions with certain objects may be less accurate when compared to using a real robot for collection, and whether the fine tune step can work as well for any arbitrary objects

- The state-of-art may solve canonical tabletop manipulation tasks soon. A more diverse task domain, e.g. mobile-manipulation for tasks beyond tabletop, would be more data hungry, and also involve novel objects which could not be easily brought into the lab (e.g. open a unique door). The proposed method will demonstrate larger impact if efficacy can be shown along those lines.



**Quality Of The Limitations Section:**

Limitations are addressed clearly

**Questions For Rebuttal:**

7: why is enabling diverse objects an benefit from the proposed system? What are the scenarios where the same novel objects can not be simply added in a traditional teleop setting?

58-60: can you provide insight into the main contributor to the sim2real gap in your setup? Does the proposed fine tune method work in all scenarios? If not, what are the limitations?

113-117: how is collision detection done? Does the system handle the following scenario: I reach into the microwave to grab an object. During the process my forearm collided with the microwave door.

223-225: are the positions of these objects randomized during the test?

255-156: `our system’s data even surpasses the real-robot collection data in tasks such as pushing the LEGO train and picking up the paper bag`: what does this mean? Intuitively, it appears that the real robot collection should be the upper bound if all else being equal. Does this indicate some other variables at play, e.g. poorer IK in real, or hardware issue? Or should this be attributed to statistical noise? Alternatively, does this actually indicate an advantage for AR2D2 in terms of *data quality*? If so, what is the main contributor to the advantage?

278: what is the number of participants who took part in the presented data collections?









**Robotics Focus:**

Sufficient demonstration on hardware

**Summary Of Paper:**

This paper presents a system which enables robot manipulation data collection using an iOS app. The authors leveraged the built-in hand gesture detection, combined with inpainting to remove the hand image, and canonical AR support to project an arm into the scene. The AR robot arm is then controlled using motion planning to track the 6DOF hand gesture. Thus, a complete demonstration including RGB, depth, arm action can be generated and collected. The authors demonstrated the effectiveness of the proposed system by training BC models using AR2D2 data and data collected with baseline methods, and evaluating in the real world. Comparable performances were shown between the tested collection methods, while the proposed method was quantitatively shown to be faster, as well as being more flexible and applicable to more environments as it requires an iOS device rather than a physical robot.

**Summary Of Recommendation:**

This paper presented an effective way to collect data in real world without requiring a real robot. The method removes one major cost for robot data collection, and therefore can contribute to significantly scaling up data collection both in quantity and environment diversity. The paper is overall well presented. However, this paper will benefit if some of the unique aspects of this setup, e.g. collision detection and its potential limitations, is discussed in more details. I would also appreciate more insight into applications in mobile-manipulation settings.

---

### Official Review · Reviewer_FwiR · 2023-07-16

**Confidence:** 4
**Originality:** Good
**Technical Quality:** Good
**Clarity Of Presentation:** Good
**Impact:** 4

**Recommendation:**

Weak Accept: I recommend accepting the paper, but will not argue for my recommendation if the majority of other reviewers have a different opinion.

**Review:**

Quality: ​​This paper presents a novel setup for learning from demonstration, which significantly improves the accessibility of robot training using real-world data. The authors effectively demonstrate the practicality of their approach through a working case study involving a smartphone application in real-world robot training tasks. The integration of these innovative ideas and technologies in this paper highlights its contribution to advancements in robotic applications.

Clarity: The paper is well-written and easy to follow. The authors have done a good job of explaining the motivation for the work, the technical details of the system, and the experimental results. They have mostly avoided using jargon and acronyms, which makes the paper accessible to a wider audience. The figures and tables are also mostly well-labeled and easy to understand.

Originality: The paper is original. The authors have developed a novel robot demonstration collection system that has the potential to make robot training more accessible. The paper also makes a good contribution by demonstrating that AR2-D2 can be used to train a real-world robot to manipulate personalized objects with comparable performance to training using real-world demonstrations.

Significance: The impact of the paper is above average. The development of AR2-D2 has the potential to significantly facilitate the way that robots are trained. It could be used to train robots to manipulate a wider variety of objects, which would advance the deployment of robots in the real world.

Strengths: The main strengths of the paper are:

- The development of a novel robot demonstration collection system that is easy to use and can be used to collect data for a wide variety of objects.
- The demonstration that AR2-D2 can be used to train a real-world robot to manipulate personalized objects with comparable performance to training using real-world demonstrations.
- The user study, which showed that users found AR2-D2 to be intuitive and easy to use.

Weaknesses: The main weaknesses of the paper are:

- The need for fine-tuning to match the performance of real data.
- Unclear explanation of the manipulation behaviors.
- The inability to validate experimental outcomes via simulation.
- The small participant pool in the user study.
- Insufficient details about the implementation of the system. This would be helpful for other researchers who want to use/replicate the approach.

Overall: I think the paper is good. The authors have made a good contribution to the field of robot learning by developing AR2-D2, a novel robot demonstration collection system that has the potential to facilitate robot training. I am excited to see how AR2-D2 is used in the future to advance the field of robot learning.


**Quality Of The Limitations Section:**

Additional details required

**Questions For Rebuttal:**

Here are some questions:

- As the main value proposition of this paper is to alleviate data collection process for robot manipulation training, fine-tuning seems to remain as a major bottleneck? Authors propose some high-level suggestions, but I’d like to hear/read more concrete solutions to make this approach really useful for the intended scenarios. How do the authors plan to address the need for fine-tuning? Is it scalable? How could this be streamlined?
  - What if users directly use the VR setup but acquire more/diverse data, would it still be justified to have two separate processes, i.e., how closer VR-only training performance can get to AR2-D2 performance?

- The visual materials and accompanying video present an apparent utilization of a person's hand for guiding the virtual robot. However, the specific mechanism by which the robot grasps and lifts objects remains unclear. It is not evident whether the person's hand physically lifts the object, followed by alignment with the robot's gripper. This lack of clarity poses a limitation to comprehending the execution of such actions. To address this issue, it is recommended to incorporate video demonstrations that provide step-by-step guidance for various tasks. The paper should also clearly explain this process. This inclusion would enhance the clarity and provide a more comprehensive understanding of the interaction between the person's hand and the robot throughout the manipulation tasks.

- How does AR2-D2 handle occlusions? What are (other) possible problems of the suggested data collection approach, esp. due to the inherent nature of the AR process?

- What are the backgrounds of the participants? How might lack of robotics knowledge affect usability and performance? How might inexperience of mobile technology, AR, etc. affect usability and performance? Since experimentation is much faster than previous tech (as the authors claim), it might be a worthy effort to extend the user study, if those issues had not been considered.

- Pushing behavior is also somewhat unclear. Does it incorporate an object specific push distance, force, etc.? Or does it just depend on a time limit?

- What’s the difference between push and press?

- I’m not familiar with RLBench, how is it different from VR training? Why does it perform so poorly (are manipulation behaviors guaranteed to succeed while collecting data)? Overall, the comparison does not say much without a clear understanding of the details (e.g., input/output, training, etc.)?

- What are the possible ideas/approaches to validate the experimental results using simulation?



**Robotics Focus:**

Sufficient demonstration on hardware

**Summary Of Paper:**

The paper introduces AR2-D2, a system for collecting human demonstrations in robot learning. Unlike existing methods that require specialized training and real robots, AR2-D2 is an iOS app that allows users to record videos of themselves manipulating objects while capturing essential data for training real robots. The key contributions of AR2-D2 are threefold: it eliminates the need for specialized training, it does not require real robots during data collection, and it enables the manipulation of diverse objects. The experiments demonstrate that training with AR2-D2 data is as effective as using real-world robot demonstrations. Additionally, a user study shows that users find AR2-D2 intuitive to use without requiring prior training, distinguishing it from other common methods. Overall, the paper presents AR2-D2 as a practical and accessible approach to collect demonstrations for robot learning, expanding the possibilities for training robots with diverse object manipulation tasks.


**Summary Of Recommendation:**

I believe that the paper is well-written, technically sound, and has the potential to have a good impact on the field of robot learning. The ability to collect demonstrations for a wide variety of objects without access to a real robot will make it possible to train robots to manipulate a wider variety of objects. I think that the paper is well-written and technically sound. The authors have done a good job of explaining the motivation for the work and the overall process. Having said that, some further clarifications, especially on the details of the AR data collection, how manipulation skills are defined, and the experimental procedures, will make the paper even better.

---

### Official Review · Reviewer_PnSQ · 2023-07-22

**Confidence:** 5
**Originality:** Excellent
**Technical Quality:** Very Good
**Clarity Of Presentation:** Very Good
**Impact:** 4

**Recommendation:**

Weak Accept: I recommend accepting the paper, but will not argue for my recommendation if the majority of other reviewers have a different opinion.

**Review:**

The paper is very well-written and the findings are crisply and convincingly demonstrated. Overall, I really like the idea of generating cheap simulation-esque data for training policies and transferring to the real setup with a small amount of real data, and this seems like a great way to get realistic data without having to construct complicated sim scenarios of all tables/objects etc. The user study is very informative and answers any doubts I had getting into this paper, and I wish more robot learning papers did this.

I do have a few concerns that I hope the authors can address:
- A fundamental limitation of the paper is the simplistic nature of tasks considered, commonly known as “pose matching”. While the findings and proposal are definitely interesting, I find the task to be of fairly limited utility, and this may be the biggest deterrent to more research labs/groups adopting this setup. What would it take to use this methodology for a more realistic task, say pick-and-place, or something more challenging, such as dexterous manipulation? While demonstrating this may be out of scope for the current paper, it is fundamental to at least address these limitations and discuss how this can be extended to more interesting/useful tasks. Otherwise, for the simple task of “reach object X”, much simpler approaches can work (e.g. have a few-shot detector or SAM pick the object and servo to it). This does not take away from the merits of the paper, but somehow feels incomplete.
- Another fundamental assumption to challenge is the dependency on end-effector control, even if the demos could be made more expressive and useful. Fundamentally defining demo trajectories with end effector positions, followed by motion planning, limits the variety of behaviors that can be demonstrated, and hence learned, by the system. This really needs to be addressed better.
- Is there any plan to open source the system design components and the iOS application developed for this purpose? Assuming the primary contribution of the paper is the validity of such an approach for data collection, it would be very beneficial for other researchers to use this directly. Otherwise, its a heavily engineered system with many moving parts, and may be a reproducibility nightmare (not more than actually dealing with robots, granted).
- Could the authors share more information on the demographics of the users studied? I can imagine the SUS scores varying greatly between CS grad students v/s STEM undergrads v/s random turkers etc. using the system, and having access to this variable seems important for readers to interpret the message.
- While not fully related, it may be good to acknowledge work in the area of third-person imitation for human-to-robot transfer of demonstrations. This may seem like a fairly related area of research (at least at the surface) and discussing the differences in assumptions and objectives would be useful.
- Small typo: L171-172 seems to reference Figure 4, rather than Figure 3.


**Quality Of The Limitations Section:**

Additional details required

**Questions For Rebuttal:**

See list above.


**Robotics Focus:**

Sufficient demonstration on hardware

**Summary Of Paper:**

The authors present a new way of collecting robot demonstrations for manipulation using an augmented reality (AR) framework, that requires no real-world robots and can be done conveniently by users in their personalized setups. The proposed methodology is compared to alternatives (such as using VR, kinesthetic demos etc.) in a user study, and is shown to be as good as expensive kinesthetic demos while being much faster to collect data, with the added benefits of customization and not requiring a real robot.


**Summary Of Recommendation:**

This is a great paper with a simple and convincing proposal. However, the authors should address the limitations of regarding the simplicity of the task and end-effector specification limiting the variety of learnable behaviors, for the system to be of use to the broader community. I really like this idea and am willing to upgrade my recommendation after further discussions in the rebuttal period.

---

### Author Response · Authors · 2023-08-07
**General Response for Rebuttal to All Reviewers & AC**

We would like to thank all the reviewers for their comprehensive and constructive feedback. It is encouraging to note the unanimous appreciation for the concept of using AR as a novel approach to collecting demonstration data. We are heartened by the overall positive reception and would like to highlight the specific commendations received for our work:

“Findings are crisply and convincingly demonstrated”, “great way to get realistic data”,”great paper with a simple and convincing proposal”, and “wish more robot learning papers did this” (Reviewer **PnSQ**)

“The paper is original”, “good impact on the field of robot learning”, “well-written and technically sound”, and “good job of explaining the motivation for the work and the overall process”  (Reviewer **FwiR**)

“Convincing results”, “effective way to collect data”, “contribute to significantly scaling up data collection ”, and “the paper is overall well presented” (Reviewer **sq69**)

“Recommends strong acceptance”,”a novel approach”,  and “enjoyable to read” (Reviewer **UipD**)

Summary of responses to the reviews (**see individual responses for more detail**):

**Evaluating AR2-D2 with more complex tasks**

To further demonstrate our method's broader capability, we experimented with the following new task - 'insert_tape_into_hanger'. To complete this task, the robot needs to not just  pick up the tape but also interact with a second object, the hanger, into which it inserts the tape. Users created 3 to 8 key-points per demonstration to complete this task. Even though this task is longer horizon and involves interacting with multiple objects, the AR2-D2 demonstrations collected train an agent to yield results consistent with the three skills already evaluated in the paper. We aim to further scale-up the diversity of tasks in future work.

**The need for additional fine-tuning data**

We appreciate the suggestions that reviewers have pointed out on the topic of fine-tuning. As suggested, we are considering the implementation of additional data augmentation techniques such as contrast adjustment, Gaussian noise, and colour jittering. While our current approach includes rotating, and translating the 3D voxel scene, we anticipate that pixel-level augmentations could potentially reconcile discrepancies between data captured by different sensors, such as iPhones/iPads and Kinect. We also anticipate that further systems such as hand removal at both pixel and depth level, and performing depth completion using the in-painted image, will reduce the need for fine-tuning efforts. Despite these considerations, our existing system, with minimal fine-tuning, has already proven its capability to collect meaningful robot demonstration data. We believe our method holds considerable potential for scaling up robot learning.

**Clarification on the demonstration collection details**

To contextualize our response, there are two types of demonstrations that can be collected with AR2-D2: video and 3D voxel demonstrations. In both cases, the human hand physically interacts with and moves a real-world object. From that demonstration, we generate both videos and 3D voxel outputs. For videos, we remove the human hand through in-painting and move the AR robot hand as an overlay. This results in a video demonstration where the robot appears to grasp and lift objects in sync with the human hand. For 3D voxels, we only record the hand pose (coupled with metadata of the 6D pose of the gripper tracking the hand pose) and the 3D voxel of the scene. We have made this more clear in our writing. We have released video demonstrations with step-by-step guidance for various tasks and the final demonstration outputs. We have also released the step-by-step guidance for collecting demonstrations on the anonymous project page: https://anonymousar2d2.github.io/AR2D2.github.io/

**Clarification on the demographics of the user-study participants**

Another major contribution of this work is to show that our method is intuitive and preferred by users.  Hence, to further address the two of the reviewers’ questions, we have added a detailed demographic graph about the participants in Figure 6 (in the supplementary material) and found that 80% of our participants are CS grad students while the other 20% are non-CS people with no technical background. Aside from their demographic information, we also report their experience with Augmented Reality (AR), familiarity with robotics, among other relevant details.

We welcome any additional questions or comments from the reviewers. We are committed to addressing them thoroughly and promptly.

---

### Decision · Program_Chairs · 2023-08-30

**Decision:**

Accept (Poster)

**Comment:**

Summary:
The paper presents a novel method of data collection for robot manipulation using augmented reality through an iOS app. The approach cleverly utilizes hand gestures and AR to simulate robot actions in real environments. The method presents an effective, scalable solution for collecting robotic data, reducing the need for physical robots.

Strength:
- Scalability: The proposed system offers a scalable way to collect robotic data without physical robots. This can enhance robot learning by providing varied real-world scenarios, without the constraints of lab environments.
- Usability and Speed: The paper demonstrates competitive model performance, and has presented convincing evidence regarding the efficiency and speed of data collection using the proposed method.
- Novelty: The usage of AR for robot data collection in real-world settings, while bypassing the need for teleoperation, is innovative and commendable.

Weakness:
- Fine-tuning with Real Robot Data: Despite the AR-based approach, the system still requires fine-tuning using real robot data. This presents a hybrid data scenario, potentially complicating the learning process.
- Fidelity and Accuracy: Concerns about the accuracy of the simulated interactions have been raised, especially when considering certain objects or situations. The readers would be keen to understand the limitations of such scenarios, and whether the gap might affect the system's performance.
- Collision Detection: Reviewers have mentioned concerns about collision detection and its potential limitations, citing a specific scenario for further clarification.
- Methodology Details: Reviewer has highlighted some missing details in the paper, particularly regarding the parametrization of gripper actions, in-painting of user’s hand in depth data, and assumptions regarding camera pose and hand occlusion.
- Small number of subjects and lack of information on whether subjects have basic knowledge of robotics

Rebuttal and Discussion:
Through the discussion in the rebuttal phase, the authors and reviewers discussed the following concerns, and the authors appropriately replied to reviewers’ questions:
- The benefit of enabling diverse objects and the limitations in traditional teleoperation settings.
- Insight into the contributors of the sim2real gap.
- How collision detection is managed and if specific scenarios are handled.
- Gripper action parametrization and depth data in-painting.
- Assumptions regarding camera pose, hand occlusion, and if bimanual motions can be accommodated.

Recommendation:
Based on the review comments and discussion, it's clear that the paper presents a promising approach. However, some concerns and questions need addressing in the final manuscript. AC recommends and believes that the authors provide an appropriate revision addressing the questions and concerns raised by the reviewers. If these issues are addressed satisfactorily, the paper holds great potential for acceptance.